# Analysis of Genetic Diversity and Population Structure in Bitter Gourd (*Momordica charantia* L.) Using Morphological and SSR Markers

**DOI:** 10.3390/plants10091860

**Published:** 2021-09-08

**Authors:** Ahmad Alhariri, Tusar Kanti Behera, Gograj Singh Jat, Mayanglambam Bilashini Devi, G. Boopalakrishnan, Nada F. Hemeda, Ayaat A. Teleb, E. Ismail, Ahmed Elkordy

**Affiliations:** 1Division of Vegetable Science, ICAR-Indian Agricultural Research Institute, Pusa Campus, New Delhi 110012, India; gbkrishnan85@gmail.com; 2Faculty of Agriculture, Damascus University, Damascus 30621, Syria; 3ICAR-Research Complex for North Eastern Hilly Region, Umiam 793103, India; bilashini1712@gmail.com; 4Department of Genetics, Faculty of Agriculture, Fayoum University, Fayoum 63511, Egypt; nfh00@fayoum.edu.eg (N.F.H.); aat01@fayoum.edu.eg (A.A.T.); 5Genetics Department, Agriculture Faculty, University of Sohag, Sohag 82524, Egypt; emad.eldeen.ismail1976@gmail.com; 6Biodiversity and Environment Management Department, Faculty of Biological and Environmental Sciences, University of Leon, 24071 Leon, Spain; aelkordy@science.sohag.edu.eg; 7Botany and Microbiology Department, Faculty of Science, University of Sohag, Sohag 82524, Egypt

**Keywords:** bitter gourd, population structure, genetic diversity, morphological traits, SSR marker

## Abstract

The present investigation was carried out using 51 diverse bitter gourd accessions as material for studying genetic diversity and relatedness using morphological and SSR markers. A wide variation was observed for morphological traits like the number of days to the first female flower anthesis (37.33–60.67), the number of days to the first fruit harvest (47.67–72.00), the number of fruits/plant (12.00–46.67), fruit length (5.00–22.23 cm), fruit diameter (1.05–6.38 cm), average fruit weight (20.71–77.67 g) and yield per plant (513.3–1976 g). Cluster analysis for 10 quantitative traits grouped the 51 accessions into 6 clusters. Out of 61 SSR primers screened, 30 were polymorphic and highly informative as a means to differentiate these accessions. Based on genotyping, a high level of genetic diversity was observed, with a total of 99 alleles. The polymorphic information content (PIC) values ranged from 0.038 for marker BG_SSR-8 to 0.721 for S-24, with an average of 0.429. The numbers of alleles ranged from 2 to 5, with an average of 3.3 alleles per locus. Gene diversity ranged from 0.04 for BG_SSR-8 to 0.76 for S-24, showing a wide variation among 51 accessions. The UPGMA cluster analysis grouped these accessions into 3 major clusters. Cluster I comprised 4 small, fruited accessions that are commercially cultivated in central and eastern India. Cluster II comprised 35 medium- to long-sized fruited accessions, which made up an abundant and diverse group. Cluster III comprised 11 long and extra-long fruited accessions. The polymorphic SSR markers of the study will be highly useful in genetic fingerprinting and mapping, and for association analysis in *Momordica* regarding several economic traits.

## 1. Introduction

Bitter gourd (*Momordica charantia* L. 2n = 2x = 22), known as bitter melon or balsam pear, is an economically and horticulturally important multipurpose vegetable of the Cucurbitaceae family. Compared with other cucurbits, it is highly valuable for its nutritional content, providing carbohydrates, proteins, vitamins, minerals, and ascorbic acid [1], and for numerous medicinal uses [2]. The juice has been used for centuries in Ayurveda as an ancient traditional medicine for treating diabetes [3] and also possesses anti-microbial, antioxidant, and anti-viral activities [4,5]. The primary center of origin for bitter gourd is probably India, with China as a secondary center of diversity [6,7]. India is richly endowed with bitter gourd genetic resources, as Indian bitter gourd germplasms are diverse in phenotypic characteristics, i.e., sex expression [8], growth habit, maturity, fruit shape, size, color and surface texture [9,10,11,12,13,14]. Nevertheless, Indian bitter gourd accessions exhibit valuable genes that are as yet unexplored [15]. Natural populations of cultivated *Momordica* species are threatened by a variety of factors in their native region. Therefore, the assessment of genetic diversity among available germplasm resources of this species is important, both for germplasm conservation and for the identification of desirable traits.

Diversity studies based on morphological characters have been extensively carried out in bitter gourds, but these are no longer sufficient because of low phenotypic polymorphism. In contrast, molecular markers would reduce the breeding cycle, and these are not influenced by environmental conditions [16], are detectable at all stages of plant growth and development [17], and show a higher level of polymorphism. The genetic diversity of bitter gourd accessions was studied using molecular marker, RAPD [18], ISSR [19], RAPD and ISSR [20,21], AFLP [22], SSR [23,24,25,26], and whole-genome re-sequencing [27]. Among molecular markers, simple sequence repeats (SSRs) are ideal genetic markers because of their multi-allelic nature and co-dominant inheritance [28], exhibit extensive genome coverage, require a small quantity of starting DNA, are easily detected by polymerase chain reaction [29,30], and are characterized by high polymorphism [31]. However, only a limited numbers of microsatellite markers are available for bitter gourds. Sixteen markers were developed using the FIASCO technique [23,32], 11 through genomic library enrichment [33], and 43 from other cucurbits through cross-species transferability [34]. A limited quantity of research has been conducted on population structure and genetic diversity in cultivated *Momordica* accessions, using SSR markers. Therefore, the present study was designed to assess the genetic diversity and population structure of 51 bitter gourd accessions of diverse origin, to identify potentially useful germplasm for its genetic improvement.

## 2. Results

### 2.1. Performance of Accessions Based on Quantitative Traits

Significant differences were observed among the 51 accessions for all 10 traits studied. The mean performance of the accessions, standard error of difference (SE), coefficient of variation (CV) and CD (*p* = 0.05) value with respect to yield; the earliness characteristics of these accessions are presented in Table 1. The results of quantitative characteristic analysis suggest that 51 accessions have unique importance since small-fruited varieties are cultivated in central and eastern India, whereas medium-long to extra-long fruited accessions are cultivated in other parts of the country, depending on consumers’ specific requirements. The gynoecious line PVGy-1 showed superior performance for earliness traits, like the number of days to the first female flower appearance (37.33 ± 2.0 days after sowing), and the node number of the first female flower appearance (4.83 ± 0.8 node), followed by another gynoecious line, PDMGy-201 (38.0 ± 2.7 days after sowing and 5.67th ± 1.1 node, respectively). The gynoecious line PDMGy-201 was also found to be superior in terms of the number of days to the first fruit harvest (47.33 ± 4.7), followed by PVGy-1 (47.67 ± 2.0). These gynoecious lines did not produce any male flowers until their last fruit harvest, which indicated the potentiality of these gynoecious lines for future breeding programs to develop F1 hybrids for better earliness characteristics. The accession DBGS-38-1 recorded the highest number of female flowers (53.33 ± 5.5), male flowers (196.00 ± 28.0), and the number of fruits per plant (46.67± 7.5). The accession Sel-2 had the maximum fruit length (22.23 ± 1.0 cm), followed by Pusa Aushadhi (19.57 ± 2.8 cm). The variety Pusa Rasdar exhibited the maximum fruit diameter (6.38 ± 0.8 cm), followed by Nakhra Local (5.75 ± 0.9 cm), whereas the variety named Pusa Aushadhi recorded the highest average fruit weight (77.67 ± 2.5 g), which was statistically at par with Pusa Vishes (77.25 ± 7.0 g). The accessions exhibiting the highest yield per plant were Sel-51 (1976.0 ± 186.5 g), Sel-57 (1736.0 ± 101.7 g), and Sel-32 (1696.0 ± 143.6 g). The gynoecious lines PVGy-1 and PDMGy-201 were observed to be early in terms of flowering, whereas Sel-51, Sel-57 and Sel-32 were promising regarding their yield and the relevant contributory traits.

### 2.2. Clustering Based on Quantitative Traits

A cluster analysis of 51 bitter gourd accessions, based on 10 quantitative characteristics, was performed using the unweighted pair group method with arithmetic mean (UPGMA) and, as a result, a dendrogram was constructed (Figure 1). All 51 accessions were grouped into six major clusters. In cluster I, the gynoecious line PDMGy-201 was quite distinct from the other 3 monoecious genotypes. In cluster II, the varieties Pusa Rasdar and Pusa Aushadhi were quite distinct from the rest of the 9 accessions, whereas, in cluster III, the variety Sel-32 was found to be very different from Sel-57 and PVGy-1. In cluster IV, the genotypes DBGS-42 and NEH-3 were found to be quite close to each other, whereas two accessions, Sel-46 and Andhra Collections, were quite distinct from the other 9 genotypes in the same cluster. The genotypes Sel-30-1, NEH-4, Arka Harit and NEH-1 were closely related in cluster IV. In cluster V, the maximum similarity was recorded between Sel-52 and Sel-27; these were distinct from the other 5 accessions, whereas Sel-25 was distinct from the other 6 accessions in cluster V. In cluster VI, the maximum similarity was observed among Sel-53, Sel-58, and Pusa Do Mausami. The variety Pusa Poorvi was found to be most dissimilar to the rest of the 50 accessions.

### 2.3. Genetic Diversity

All the 51 accessions were screened using 61 SSR markers, but only 30 were polymorphic and produced a total of 99 alleles. The number of alleles per locus varied from 2 to 5, with an average of 3.3 alleles per locus. The highest number of alleles (5) was detected for the loci AVRDC BG-66, JY003, S-24 and S-32, and the lowest (2) in markers AVRDC BG-2, AVRDC BG-74, BG_SSR-8, McSSR-20 and McSSR-22 (Table 2). The allele frequency of these accessions varied from 0.35 at JY001 to 0.98 at BG_SSR-8, with a mean of 0.65 at each locus. The frequencies of alleles were low, particularly for the loci with the higher number of alleles. The PIC value represents the relative information of each marker and, in the present study, the average polymorphic information content value was found to be 0.429, which ranged from 0.038 for BG_SSR-8, to 0.721 for S-24 (Table 2). The observed heterozygosity (H_O_) varied from 0.00 for AVRDC BG-1, BG_SSR-8, AVRDC BG-95, McSSR-20, JY004, JY006, N6, N9, JY005, S26 and S20, to 0.24 for McSSR-22 (Table 2). The number of accessions obtained per locus ranged from 2, for BG_SSR-8, McSSR-20, and McSSR-20, to 8 for S-24.

### 2.4. Genetic Structure Analysis

The population structure of the 51 accessions was analyzed using a Bayesian-based approach. The estimated membership fractions of 51 accessions for different values of K ranged between 3 and 51. The log-likelihood, revealed by structure, showed the optimum value as 3 (K = 3). Similarly, the maximum of ad hoc measure for K was found to be K = 3 (Figure 2). Based on the ΔK method, the results suggest that the 51 bitter gourd accessions can be grouped into three sub-groups. On the basis of membership fraction, those accessions with a probability of ≥ 60% were assigned to subgroups, with another category as admixture. Clustering analysis based on the UPGMA method separated the accessions into three groups, which showed similar results as in the STRUCTURE analysis. The UPGMA cluster analysis showed that all 51 bitter gourd accessions could be clustered into 3 groups (Figure 3). Group I consisted of 4 small-fruited bitter gourd accessions that are commercially grown in central and eastern India. Group II consisted of 35 accessions and was the most genetically diverse group, comprising medium-sized fruited accessions of diverse origin, including exotic lines. Group III comprised 11 accessions with long and extra-long fruited accessions. The accession Sel-25 was found to be more diverse than the rest of the 50 accessions, on the basis of genotyping and analysis using clustering and the UPGMA tree. Principal component analysis (PCA) data for all 51 accessions are shown in Figure 4. The analysis classified these accessions into different groups involving different bitter gourd accessions of diverse origin. Among these, the accessions of four groups were highly diverse from the rest of the accessions. The accessions of these four groups are: Group I (Arka Harit, Sel-52 and Nakhra Local), Group II (Sel-25, Sel-41, and NEH-1), Group III (Sel-52-1, Sel-52-2, Sel-52-3, Sel-33, and Sel-46), and Group IV (Sel-53, Sel-37, Sel-42, Sel-26, Sel-33-2, and Pusa Aushadhi). The amount of variance accounted for by the PCA plot is 13.48% of axis 1, 7.23% of axis 2, and 6.76% of axis 3, with a total of 27.47% for the three axes. This is an acceptable fit, given the substantial amount of variability from the 51 accessions and SSR alleles used in the analysis.

## 3. Discussion

A population with a high level of genetic diversity is an important resource for the expansion of a genetic base in any crop-breeding program. The group of 51 bitter gourd accessions in this study, including released varieties from India as well as breeding lines, has diverse prominent agronomic traits that are of economic importance. Several accessions included in this study have extra-small or extra-large fruits, available according to consumer-specific preferences in certain regions. Two gynoecious lines, PVGy-1 and PDMGy-201, one predominantly gynoecious variety Pusa Aushadhi, and one monoecious variety, Pusa Rasdar, are an important resource for early flowering and fruiting, but few monoecious accessions have significant yield potential. These accessions showed a wide morphological variation with regard to a number of traits, including flowering time, fruit shape, size, color, and yield variation. In the bitter gourd germplasm, maximum diversity was observed for numerous characteristics, including sex expression (i.e., gynoecious, predominantly gynoecious, or monoecious [8]), growth habit, maturity, fruit shape, size, color and surface texture [9,10,13,14,26]. Line DBGS-38-1 recorded the highest number of fruits per plant, which could be due to the presence of the maximum numbers of female and male flowers (desirable ratio of female to male flowers) during the whole growing season. Line Sel-2 was promising in terms of maximum fruit length, and Pusa Rasdar for fruit diameter. Sel-51, Sel-57 and Sel-32 were high-yielding accessions. These accessions, including the gynoecious lines PVGy-1 and PDMGy-201, could be used in future breeding programs for the development of early- and high-yielding gynoecious bitter gourd varieties, along with hybrids showing tuberculation and dark green fruits. The potentiality of the gynoecious lines for earliness traits and increased yield in bitter gourd was also reported [8,13,18,20,22,35]. The cluster analysis, based on ten quantitative characters, showed that the line PDMGy 201 was distinct from the rest of the three genotypes in cluster I, which could be due to its unique characteristics, like gynoecious reproduction, and its superiority for earliness traits like the number of days until the first female flower appearance and the number of days until the first fruit is harvested. The variety Pusa Rasdar was also diverse from the remaining 9 accessions in cluster II; this may be due to its unique fruit characteristics, such as a capsicum-shaped fruit, larger fruit size, and its greatest fruit diameter in comparison to the other accessions studied. The variety Pusa Rasdar was genetically closer to Pusa Aushadhi in the same cluster, which may be due to their predominantly gynoecious reproduction, which promotes early flowering and fruiting. The variety Pusa Poorvi was highly distinct from all 50 accessions studied, which may be due to wild bitter gourd species (*M. charantia* var. *muricata*) and specific characteristics like small fruits (average weight is 20.71 ± 1.5 g), minimum fruit length (5.00 ± 0.4 cm), minimum fruit diameter (1.05 ± 0.3) and the highest number of fruits (42.62 ± 2.7/plant).

Molecular markers play an important role in identifying the level of genetic diversity among traditional races, released varieties and exotic lines [22]. The genetic diversity of Indian and exotic bitter gourd lines had been precisely estimated using molecular markers, i.e., RAPD [18], ISSR [13,19], RAPD and ISSR [20,21], AFLP [22] and SSR [24]. In this study, the genetic diversity among the 51 accessions was evaluated by model-based clustering and distance-based clustering approaches, using the SSR genotypic data. Thirty polymorphic SSR markers have been detected, giving a total of 99 alleles across 51 accessions (Table 2). The number of alleles varied from 2 to 5, and the average was 3.3 alleles per locus. The observed heterozygosity ranged from 0.02 to 0.24. Saxena et al. [24] recorded 68 alleles using 51 SSR markers, and the number of alleles per locus ranged from 2 to 5 (on average, 2.80 alleles per locus). Saxena et al. [24] estimated that the PIC value ranged from 0.139 to 0.775 (average 0.369); the observed heterozygosity ranged from 0.042 to 0.587 (average 0.106) among 54 accessions of bitter gourd. The SSR markers used for genetic diversity in this study were different to those used in our study. Saxena et al. [24] used 54 bitter gourd accessions that were quite different and represented only a few states in India; they were mostly exotic collections but in our study, accessions of diverse origin were used. These accessions include most of the gynoecious lines and commercially cultivated varieties from India. Therefore, the polymorphic SSR markers of the present study will be highly useful in fingerprinting, genetic mapping and association analysis of bitter gourd varieties for several economic traits. Guo et al. [23] recorded the number of alleles per locus, which ranged from 2 to 5 (average 3.1) for the Luoyang population and 2 to 5 (average 2.6) for that in Guangzhou. On average, the observed heterozygosities were 0.31–0.78 (average 0.52) and 0.12–0.75 (average 0.47) for the Luoyang and Guangzhou populations, respectively [23]. Out of 25 SSR markers screened, 6 were polymorphic, with a PIC value ranging from 0.22–0.94 [26]. In the present study, the average number of alleles (3.3 alleles/locus) was slightly higher than the average number of alleles (2.80 alleles/ locus) as reported by Saxena et al. [24] in 54 accessions of bitter gourds, with 3.1 alleles per locus in Luoyang and 2.6 alleles per locus in Guangzhou populations as reported by Guo et al. [23]. Using 54 germplasm lines, Saxena et al. [24] reported a total of 68 alleles for all loci, with a mean of 2.80 alleles per locus and a mean PIC of 0.369 when screening with 51 SSR markers. The gene diversity detected in the present study (0.51) of 51 accessions represents a large proportion of the genetic diversity that exists in major bitter-gourd-growing in the Asian continent. The PIC value was 0.429, which varied from 0.038 for the locus BG_SSR-8, with only two alleles, to 0.721 for S-24, which allowed the amplification of 5 alleles. These values were higher than those found by Saxena et al. [24] in 54 accessions of bitter gourd, with a polymorphic information content of 0.139–0.775 (average 0.369). Kumar et al. [26] reported a PIC value ranging from 0.22–0.94, whereas Dalamu et al. [22] found 0.17 and 0.40 PIC values using 11 ISSR and 17 RAPD markers in 50 accessions of bitter gourd, respectively. In the study by Dalamu et al. [22], a considerable number of rare alleles was identified, which indicated that these rare alleles contributed well to the overall genetic diversity of the population. In the current study using a bitter gourd diversity group of 51 accessions, based on the criterion of maximum membership probabilities, 4 accessions were assigned to the major Group I, which is dominated by the small-fruited type of bitter gourd varieties of Indian origin that are commercially grown in central and eastern India, and Group II consisted of 35 accessions. This was the most genetically diverse group, with medium-long fruited and exotic lines. Group III comprised 11 accessions, including long and extra-long fruited cultivars. The accession Sel-25 was separated genetically from the other 50 accessions studied, which may be due to its unique characteristics, like its dark green, small ovate fruits with discontinuous ridges, sharp tubercles, a short vine, and early characteristics (days to first female flower of anthesis, node number of first female flower) compared with many monoecious accessions.

There are only a few reports available on population structure using the model-based approach in the bitter gourd [25] and cucurbit species [36]. The results of this study indicated three distinct groups, which may be due to the different adaptation behavior of various accessions to different ecological environments, since these three groups of accessions had an independent evolutionary frame, with India as the primary center of origin [8]. This study also clarified the relationship between Indian germplasm and exotic accessions, which indicated that germplasm lines vary based on ecology, and it also showed that a higher level of genetic diversity exists within these populations. Hence, the main criterion for population structure among 51 accessions is the fruit characteristics in the different regions. The results of the model-based analysis are in accordance with the clustering pattern of the UPGMA tree and PCA. The determination of population structure, followed by sampling based on the relatedness of the accessions in the population, can be executed for the establishment of an association panel in *Momordica* that can be used in genome-wide or candidate gene-specific association mapping for linking phenotyping and genotyping variations.

## 4. Materials and Methods

### 4.1. Plant Material

Fifty-one morphologically and geographically diverse *M. Charantia* accessions, including the commercially released varieties in India, two promising gynoecious lines (PVGy-1 and PDMGy-201), and exotic lines (Table 3), were used for the present experiment. These accessions were maintained at the research farm of the Division of Vegetable Science, ICAR–Indian Agricultural Research Institute (IARI), New Delhi. All accessions that were evaluated in the current study were self-pollinated six times before evaluation. Fresh leaf samples were collected from each individual accession, and each leaf sample was placed in aluminum foil in a separate labeled zip-lock polyethylene bag in an airtight container with ice chips and kept in a deep-freezer at −80 °C until being used for DNA extraction.

### 4.2. Field Evaluation and Data Collection

The phenotypic evaluation was carried out at the Research Farm of the Division of Vegetable Science, ICAR-IARI, New Delhi during spring and summer (February–May) of 2016 and 2017 in a randomized block design (RBD) with three replications. The sowing of the seeds was performed on both inner sides of the channel, maintaining a spacing of 2 m between channels and 60 cm between plants, with 90-cm wide channels for irrigation purposes. The recommended dose of fertilizer and culture practices, along with plant protection measures, were followed to raise a healthy crop. The border plants at both ends of the channels were discarded and ten plants per accession were randomly selected for phenotypic observations and were evaluated for 10 important quantitative traits (Table 1). For taking observations related to fruit traits, the fruits were harvested at the marketable stage (approximately 8–12 days after pollination). The data for ten quantitative characters were recorded during the spring and summer seasons of both 2016 and 2017, with three replications. The pooled data for each characteristic were subjected to an analysis of variance [37].

### 4.3. DNA Extraction and Quantification

The total genomic DNA was extracted from the individual young leaf tissue of 51 accessions using a modified cetyl trimethyl ammonium bromide (CTAB) DNA isolation protocol [18]. Agarose gel electrophoresis (0.8% agarose) was used for DNA quantification and quality analysis. For comparative testing of the banding morphotypes, uncut λ DNA of known quantity (50 ng) was used as a standard and was compared. The DNA was diluted in molecular-grade water or the appropriate amount of TE (Tris-EDTA) buffer to a final concentration of 25 ng/μL and then stored at 4 °C. The diluted concentration of DNA was subjected to polymerase chain reaction (PCR) amplification, using 61 SSR markers.

### 4.4. SSR Array and Genotyping

Each microsatellite PCR amplification reaction was performed with 10 µL reaction volume, containing 2 µL of 25 ng genomic DNA, 1.0 µL of 10 mM each of deoxynucleotide triphosphates (dNTPs), 1 µL of each of the reverse and forward primers, 2 µL 1X PCR standard Taq buffer (Tris with 15 mM Mgcl2) and 0.2 µL of Taq DNA polymerase (0.3 U/µL) (Bangalore GeNei Pvt. Ltd., Bengaluru, India), and 2.8µL molecular-grade water. The PCR mixture was prepared at 0 °C using ice-cooling plates and was transferred to the thermal cycler. Amplifications were performed using the Eppendorf thermal cycler (Eppendorf AG) at 94 °C for 4 min, with 35 cycles of denaturation at 94 °C for 1 min and primer annealing at 50–58 °C for 1 min, depending on the primer annealing temperature and an extension at 72 °C for 1 min, followed by a final extension of 10 min at 72 °C and finally at 4 °C. The amplified products were resolved by electrophoresis on 3% of SFR agarose gel run in 1 x TAE buffer. A standard marker (50 bp) (Bangalore GeNei Pvt. Ltd., Bengaluru, India) was added to every gel as a control. The PCR products were stored at 4 °C until the scoring of banding patterns. The allele’s length was compared with the standard bands of the standard marker and scored. The DNA samples of 51 accessions were genotyped using 61 SSR markers, and 30 markers were detected as polymorphic markers. The list of 30 polymorphic SSR markers with gene diversity parameters, like the number of alleles per locus (Ao), major allele frequency (Fx), genetic diversity/expected heterozygosity (He), observed heterozygosity (Ho), polymorphism information content (PIC), and fixation index (F) is given in Table 2.

### 4.5. Allele Scoring

The allele score was given based on the presence of a specifically sized allele in each accession. The presence of a particular band was denoted as 1 and the absence of an allele as 0, and this was rechecked manually.

### 4.6. Data Analysis

Based on the presence or absence of an allele, a 1/0 matrix was constructed for a set of 30 polymorphic SSR markers. These SSR genotyped data were analyzed for the assessment of genetic diversity for each locus, and the population structure among 51 bitter gourd accessions. Using Mahalanobis D^2^ analysis; the accessions were grouped into various cluster groupings following Torcher’s method [38]. NTSYS-pc Version 2.02 (Numerical Taxonomic System) software was used to calculate Jaccard’s similarity coefficients between accessions. For a set of 51 bitter gourd accessions, genetic diversity parameters such as the number of alleles per locus (Ao), allele frequency (Fx), gene diversity/expected heterozygosity (He), observed heterozygosity (Ho) number of accessions, polymorphic information content (PIC) value, and fixation index (F) were estimated, using the program Power marker ver. 3.25 [39]. Allele frequency represents the frequency of a particular allele for each marker. Heterozygosity is the percentage of heterozygous individuals in the population. Polymorphic information content (PIC) corresponding to the amount of polymorphism within a population was estimated [40].

### 4.7. Population Structure and Genetic Diversity Analysis

For the analysis of genetic structure, model- and distance-based approaches were followed. The model-based approach was utilized with Structure ver. 2.3.4 software [41]. The actual number of sub-populations, as denoted by K, was identified using this method. The project was run with the following parameter set, and the possibility of admixture and allele frequency were correlated. Run length was specified as a 150,000 burning-period length, followed by 150,000 Markov Chain Monte Carlo (MCMC) replication. The optimum K value was determined by plotting the mean estimate of the log posterior probability of the data L (K) against the given K value. The K value was based on the run with the highest likelihood. The software Numerical Taxonomy and Multivariate Analysis (NTSYS, https://ntsyspc.software.informer.com/2.2/ (accessed on 20 October 2018)) was applied to construct the UPGMA tree [42] on the basis of similarity measures. A neighbor-joining (NJ) method is used for reconstructing phylogenetic trees from evolutionary distance data [43]. The genetic distance between accessions was estimated using the NEI coefficient [44] with a bootstrap procedure of re-sampling (1000) across markers and individuals from allele frequencies. To determine the association among the accessions. Principal coordinate analysis (PCoA) of the 51-accession set was performed based on the Nei [45] distance matrix, using Gen AlEx 6.5 [46]. The SSR markers used in the present study include the McSSR series [24], AVRDC-BG series [47], JY series [33], N and S series [32], A and C series [23], MC series [25], and BG_SSR series.

## 5. Conclusions

The results are based on ten important quantitative traits to assess the potentiality of bitter gourd gynoecious lines regarding yield and earliness; the genetically divergent accessions identified in the present experiment will be useful in future breeding programs for enhancing the productivity of bitter gourd. This study analyzed the pattern of genetic divergence existing in 51 bitter gourd accessions; these constituted the bitter gourd diversity group for future association mapping. This SSR marker-based study has also identified the existence of a broad genetic base in the available germplasm lines of bitter gourd. Thus, the present results of genetic diversity among 51 accessions can be utilized to enhance other types of breeding analysis, such as association mapping, and trait-specific breeding for bitter gourd crop improvement programs, along with the development of hybrids for better exploitation of the natural genetic variations that exist within the available germplasm of cultivated bitter gourd.

## Figures and Tables

**Figure 1 plants-10-01860-f001:**
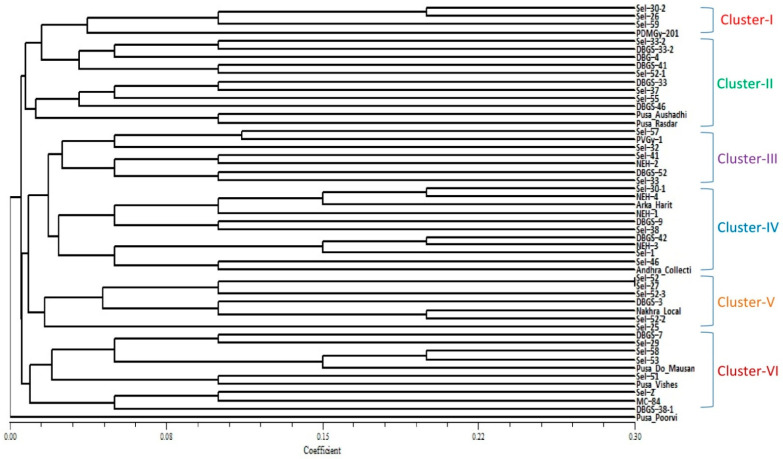
Morphological relationships among 51 accessions of bitter gourd, based on 10 quantitative characteristics by UPGMA clustering, based on Jaccard’s similarity coefficient.

**Figure 2 plants-10-01860-f002:**
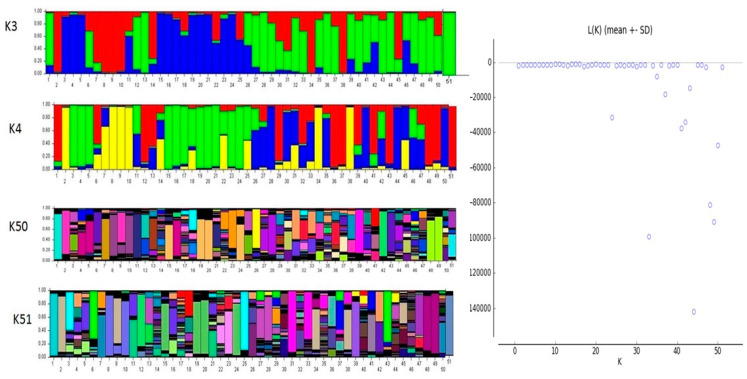
STRUCTURE analysis of Momordica accessions, based on the SSR data. Bars represent the membership coefficients of individual plants based on SSR allele frequencies, using K-values for three, four, fifty, or fifty-one groups. Numbers on the horizontal axes correspond to the population numbers in Table 1.

**Figure 3 plants-10-01860-f003:**
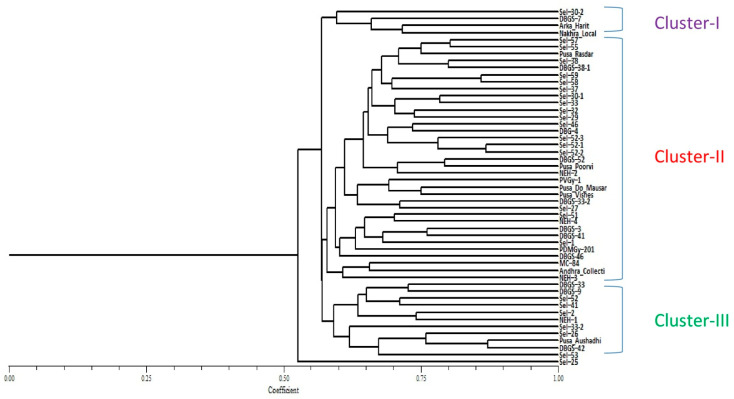
The unweighted pair group method of arithmetic mean (UPGMA) trees of 51 *Momordica* accessions, based on 30 SSR polymorphic markers.

**Figure 4 plants-10-01860-f004:**
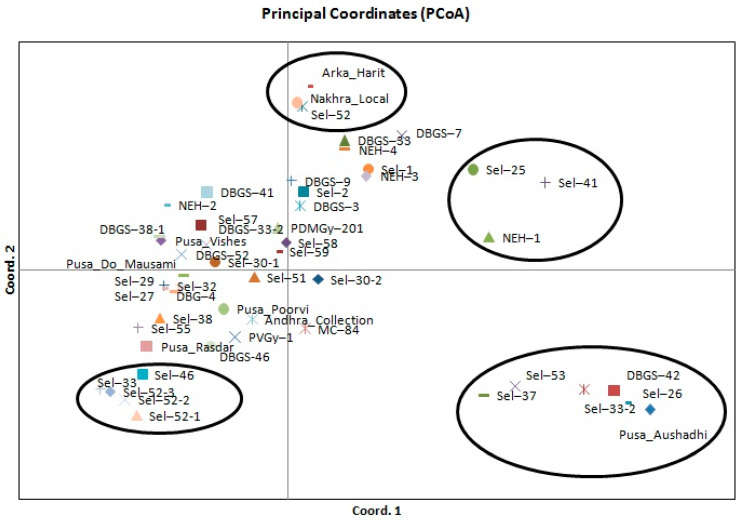
Principal coordinate analysis (PCoA); variation on axis 1: 13.48%, axis 2: 7.23%, axis 3: 6.76% and total variation 27.47%.

**Table 1 plants-10-01860-t001:** Performance of 51 accessions of *Momordica charantia* in terms of earliness and yield characteristics.

Succ. No.	Accession Name	DFFFA	NNFFF	DFFH	NFF/P	NMF/P	NF/P	FL (cm)	FD (cm)	AFW (g)	Y/P (g)
1.	Sel-30-2	54.67 ± 0.6	21.00 ± 1.0	64.33 ± 3.0	29.33 ± 2.0	106.33 ± 5.5	23.33 ± 2.5	8.23 ± 1.0	3.63 ± 0.8	38.67 ± 6.5	891.3 ± 56.7
2.	Sel-57	50.67 ± 0.6	13.67 ± 1.5	63.67 ± 1.1	33.00 ± 2.7	123.33 ± 11.4	27.00 ± 1.0	16.23 ± 0.6	4.89 ± 0.4	62.00 ± 2.0	1736.0 ± 101.7
3.	DBGS-33	48.67 ± 1.2	23.00 ± 5.6	59.33 ± 4.5	25.00 ± 3.6	97.67 ± 10.5	23.67 ± 4.5	10.67 ± 0.4	4.14 ± 1.0	39.00 ± 1.7	923.3 ± 178.9
4.	DBGS-7	59.00 ± 1.0	18.33 ± 3.8	72.00 ± 3.0	23.33 ± 6.5	88.33 ± 17.9	17.67 ± 5.5	8.13 ± 1.0	4.88 ± 1.5	44.67 ± 4.5	772.7 ± 169.1
5.	Sel-52	51.00 ± 2.0	18.67 ± 0.6	63.00 ± 4.7	21.00 ± 4.4	81.33 ± 10.7	15.33 ± 3.5	12.40 ± 0.8	4.88 ± 3.0	47.00 ± 4.6	710.0 ± 93.4
6.	Sel-30-1	55.33 ± 1.5	19.33 ± 1.5	66.33 ± 2.5	25.33 ± 3.2	92.33 ± 16.0	20.67 ± 3.0	9.87 ± 0.3	3.78 ± 0.2	33.33 ± 4.2	684.0 ± 84.0
7.	Sel-32	47.33 ± 2.0	13.67 ± 1.5	58.00 ± 2.0	27.67 ± 2.0	95.33 ± 5.0	20.67 ± 3.0	13.32 ± 0.7	4.78 ± 1.0	70.46 ± 6.1	1696.0 ± 143.6
8.	Sel-59	53.33 ± 1.5	15.33 ± 1.1	71.33 ± 1.5	30.67 ± 1.1	117.67 ± 6.5	23.33 ± 1.1	18.00 ± 1.0	4.93 ± 1.3	71.33 ± 1.5	1663.3 ± 51.0
9.	Sel-37	48.67 ± 2.5	13.67 ± 1.5	60.00 ± 2.0	25.67 ± 1.5	88.00 ± 7.5	19.33 ± 0.6	18.33 ± 0.6	5.15 ± 0.8	71.67 ± 5.5	1385.7 ± 114.6
10.	Sel-58	45.67 ± 3.2	15.67 ± 3.5	57.00 ± 4.7	23.33 ± 5.1	94.00 ± 10.8	18.33 ± 4.7	13.57 ± 0.9	5.41 ± 1.0	52.00 ± 3.0	947.3 ± 173.8
11.	Sel-2	53.00 ± 1.7	17.00 ± 2.7	64.67 ± 2.5	31.33 ± 2.3	108.33 ± 8.5	26.67 ± 2.0	22.23 ± 1.0	3.75 ± 0.7	55.36 ± 3.5	1474.9 ± 122.8
12.	Sel-51	53.67 ± 1.2	18.00 ± 1.0	63.33 ± 2.0	34.00 ± 4.6	122.67 ± 14.5	28.33 ± 1.5	18.83 ± 0.8	5.46 ± 0.8	69.67 ± 3.5	1976.0 ± 186.5
13.	PVGy-1	37.33 ± 2.0	4.83 ± 0.8	47.67 ± 2.0	42.67 ± 5.7	0.00 ± 0.0	27.00 ± 1.0	10.64 ± 0.4	4.73 ± 0.8	52.05 ± 1.8	1406.3 ± 93.4
14.	Sel-33-2	50.00 ± 4.6	17.00 ± 2.7	61.33 ± 7.0	28.67 ± 3.5	106.33 ± 12.6	21.67 ± 3.0	7.77 ± 0.6	2.96 ± 0.6	45.00 ± 4.0	983.0 ± 225.4
15.	Sel-25	47.67 ± 3.2	17.33 ± 1.5	61.00 ± 3.0	41.33 ± 1.5	128.67 ± 21.2	36.67 ± 2.0	6.57 ± 0.3	3.46 ± 0.6	29.67 ± 2.0	1090.7 ± 140.0
16.	Sel-41	50.00 ± 1.7	20.00 ± 2.7	62.33 ± 4.5	27.00 ± 3.6	109.00 ± 8.9	19.00 ± 2.7	12.27 ± 1.0	4.49 ± 1.4	36.67 ± 2.5	697.0 ± 114.5
17.	Sel-26	52.00 ± 2.0	20.33 ± 3.5	64.33 ± 1.1	31.67 ± 3.8	117.00 ± 21.4	23.33 ± 2.0	12.53 ± 0.7	3.25 ± 0.2	45.33 ± 3.5	1054.0 ± 62.4
18.	NEH-4	52.33 ± 1.1	18.00 ± 1.0	65.33 ± 1.1	25.33 ± 0.6	101.67 ± 3.2	20.67 ± 2.5	12.63 ± 0.6	3.45 ± 0.4	45.33 ± 3.5	942.7 ± 185.5
19.	Pusa Aushadhi	42.33 ± 1.2	10.67 ± 2.5	53.67 ± 1.5	33.67 ± 4.0	14.67 ± 3.5	21.67 ± 2.0	19.57 ± 2.8	4.67 ± 2.3	77.67 ± 2.5	1684.0 ± 186.0
20.	DBGS-42	52.33 ± 1.2	20.00 ± 3.5	64.33 ± 2.5	29.00 ± 4.4	104.00 ± 7.2	25.33 ± 4.0	10.30 ± 1.0	3.70 ± 1.1	42.67 ± 3.2	1089.4 ± 247.6
21.	NEH-1	51.33 ± 2.5	19.00 ± 2.6	62.33 ± 2.5	25.33 ± 1.1	109.33 ± 10.0	20.00 ± 3.6	10.60 ± 1.2	3.76 ± 1.4	43.67 ± 4.7	868.3 ± 134.4
22.	Sel-53	44.00 ± 1.0	22.33 ± 4.0	57.33 ± 2.0	23.33 ± 4.9	91.00 ± 15.6	18.33 ± 2.1	10.45 ± 0.8	4.94 ± 1.0	47.67 ± 2.5	871.7 ± 81.3
23.	DBGS-3	58.00 ± 3.0	26.00 ± 4.4	69.33 ± 4.0	19.67 ± 4.5	75.33 ± 8.0	12.00 ± 2.0	17.20 ± 3.2	4.89 ± 0.5	43.00 ± 2.7	513.3 ± 64.3
24.	Sel-1	52.33 ± 1.2	16.67 ± 1.1	63.67 ± 1.5	36.67 ± 1.2	131.00 ± 12.5	30.00 ± 2.0	10.96 ± 0.3	4.30 ± 0.8	52.36 ± 3.2	1567.6 ± 73.3
25.	DBGS-9	49.67 ± 0.6	21.67 ± 2.5	59.33 ± 1.5	24.67 ± 2.0	100.33 ± 5.5	20.00 ± 3.6	13.53 ± 0.8	5.25 ± 1.3	46.00 ± 3.6	912.7 ± 105.7
26.	Arka Harit	49.67 ± 2.0	18.00 ± 1.0	61.33 ± 1.5	25.33 ± 3.0	94.00 ± 12.8	19.67 ± 5.7	14.33 ± 0.6	3.35 ± 0.5	41.20 ± 1.0	813.6 ± 251.9
27.	Pusa Do Mausami	47.33 ± 1.5	15.67 ± 1.5	60.33 ± 1.5	29.00 ± 2.7	101.00 ± 3.6	18.33 ± 3.0	15.50 ± 1.3	4.98 ± 1.5	77.10 ± 5.8	1403.6 ± 163.6
28.	Pusa Vishes	50.00 ± 1.0	15.67 ± 0.6	63.33 ± 1.2	23.00 ± 5.6	91.33 ± 20.4	15.67 ± 5.9	13.33 ± 0.6	5.13 ± 0.4	77.25 ± 7.0	1183.5 ± 368.0
29.	Sel-46	54.00 ± 1.7	20.00 ± 1.7	67.00 ± 2.0	21.33 ± 2.3	84.33 ± 7.6	17.67 ± 1.5	13.06 ± 1.5	5.18 ± 1.0	55.33 ± 3.5	974.0 ± 26.0
30.	Sel-38	49.67 ± 2.1	16.67 ± 1.5	62.33 ± 1.5	40.33 ± 3.0	139.33 ± 12.0	33.33 ± 4.2	12.20 ± 0.9	3.60 ± 0.6	40.00 ± 1.0	1335.3 ± 194.3
31.	DBGS-52	50.67 ± 2.0	20.67 ± 3.0	61.67 ± 2.5	20.67 ± 1.5	88.67 ± 9.3	15.00 ± 2.6	7.53 ± 1.3	3.21 ± 1.1	54.33 ± 4.7	810.7 ± 130.0
32.	MC-84	43.33 ± 0.6	18.67 ± 2.5	52.00 ± 1.0	27.33 ± 2.9	108.33 ± 8.5	23.67 ± 1.5	14.67 ± 0.6	3.76 ± 0.3	52.20 ± 1.0	1234.9 ± 73.6
33.	Pusa Poorvi	45.33 ± 1.5	23.58 ± 2.7	60.75 ± 7.2	49.00 ± 1.0	192.00 ± 21.6	42.62 ± 2.7	5.00 ± 0.4	1.05 ± 0.3	20.71 ± 1.5	884.5 ± 105.6
34.	Sel-55	42.33 ± 1.6	16.00 ± 1.0	58.67 ± 0.6	20.33 ± 3.5	66.67 ± 7.8	15.00 ± 2.0	9.90 ± 0.4	5.15 ± 0.4	45.00 ± 3.6	673.7 ± 88.7
35.	NEH-2	50.67 ± 2.0	19.67 ± 2.0	62.00 ± 2.0	24.67 ± 1.5	94.33 ± 10.6	19.00 ± 2.0	11.53 ± 0.9	4.33 ± 1.3	47.67 ± 2.5	902.3 ± 48.2
36.	DBGS-4	60.67 ± 3.8	22.67 ± 3.5	69.00 ± 5.0	26.33 ± 2.0	116.00 ± 13.2	21.33 ± 3.8	11.13 ± 1.3	4.34 ± 0.4	42.33 ± 4.2	909.3 ± 218.8
37.	S-52-3	55.33 ± 3.5	19.67 ± 1.5	67.67 ± 3.0	20.67 ± 2.9	95.00 ± 12.4	15.33 ± 3.2	8.70 ± 0.7	5.54 ± 0.9	37.33 ± 1.5	571.3 ± 114.0
38.	Pusa Rasdar	41.33 ± 1.5	10.67 ± 0.6	54.27 ± 2.9	24.33 ± 1.5	74.00 ± 7.8	19.36 ± 0.8	9.43 ± 0.9	6.38 ± 0.8	65.39 ± 2.6	1267.0 ± 86.3
39.	PDMGy-201	38.00 ± 2.7	5.67 ± 1.1	47.33 ± 4.7	42.00 ± 4.4	0.00 ± 0.0	30.00 ± 3.6	10.31 ± 0.3	5.06 ± 0.6	45.33 ± 4.7	1359.7 ± 223.0
40.	DBGS-33-2	51.00 ± 4.6	23.33 ± 3.0	61.33 ± 2.5	22.33 ± 5.9	92.67 ± 21.8	18.00 ± 5.2	10.63 ± 0.7	4.34 ± 0.3	40.67 ± 3.0	722.6 ± 164.4
41.	Andhra Collection	54.00 ± 3.0	16.67 ± 2.0	65.00 ± 1.0	31.00 ± 1.7	123.67 ± 19.8	24.67 ± 1.5	14.87 ± 0.8	3.76 ± 0.8	58.57 ± 4.8	1449.5 ± 204.7
42.	Nakhra Local	51.00 ± 1.7	17.33 ± 3.0	63.00 ± 2.0	19.67 ± 1.5	80.33 ± 5.1	14.00 ± 2.6	12.33 ± 1.5	5.75 ± 0.9	41.00 ± 4.4	579.8 ± 159.0
43.	Sel-33	50.67 ± 2.3	19.00 ± 2.0	59.33 ± 4.5	22.67 ± 2.0	90.00 ± 9.5	17.33 ± 2.0	10.47 ± 0.8	5.09 ± 1.0	40.00 ± 3.0	697.3 ± 132.5
44.	Sel-27	51.00 ± 1.7	16.67 ± 2.0	62.00 ± 1.0	21.00 ± 4.4	75.33 ± 19.7	15.33 ± 3.2	10.53 ± 0.8	3.91 ± 0.6	44.00 ± 5.0	664.7 ± 66.9
45.	DBGS-38-1	52.00 ± 4.3	18.67 ± 2.5	60.33 ± 3.2	53.33 ± 5.5	196.00 ± 28.0	46.67 ± 7.5	6.30 ± 0.6	3.65 ± 0.3	31.67 ± 4.0	1457.7 ± 54.0
46.	NEH-3	52.33 ± 2.0	16.67 ± 2.3	63.00 ± 2.7	24.00 ± 2.0	89.00 ± 7.9	19.00 ± 2.0	12.17 ± 0.8	4.77 ± 0.4	42.67 ± 23.	810.7 ± 98.3
47.	DBGS-41	54.67 ± 3.0	23.00 ± 3.6	66.67 ± 2.5	29.00 ± 3.0	117.00 ± 14.0	22.00 ± 5.2	9.90 ± 0.5	4.81 ± 0.2	40.67 ± 4.6	878.7 ± 129.2
48.	Sel-52-1	57.00 ± 6.2	14.67 ± 2.5	68.67 ± 3.5	19.33 ± 3.2	86.00 ± 9.6	13.67 ± 1.5	9.87 ± 1.3	5.04 ± 1.5	40.67 ± 1.5	554.3 ± 43.9
49.	Sel-52-2	53.33 ± 6.7	14.33 ± 1.5	66.67 ± 4.7	19.67 ± 1.2	90.00 ± 9.5	12.67 ± 1.5	10.34 ± 0.8	4.76 ± 0.6	41.00 ± 3.6	515.7 ± 18.9
50.	Sel-29	47.00 ± 4.6	23.00 ± 2.0	60.33 ± 4.2	22.33 ± 2.5	92.33 ± 6.0	16.00 ± 2.0	8.13 ± 1.1	4.23 ± 0.8	41.67 ± 2.0	664.0 ± 52.4
51.	DBGS-46	48.33 ± 4.0	26.33 ± 2.5	60.67 ± 3.8	32.00 ± 5.2	121.67 ± 17.2	26.33 ± 2.5	9.53 ± 0.6	4.14 ± 1.0	33.67 ± 3.2	882.0 ± 49.1
	Mean	50.22	17.92	61.95	28.03	98.09	21.80	11.90	4.60	48.23	1028.06
	CD at 5%	4.87	4.46	5.68	5.89	22.37	5.69	1.89	0.63	6.73	260.74
	CV (%)	5.33	13.69	5.04	11.55	12.54	14.35	8.71	7.56	7.68	13.94

DFFFA = Days to first female flower appearance, NNFFF = node number of the first female flower, DFFH = days to the first fruit harvest, NFF/P = number of female flowers per plant, NMF/P = number of male flowers per plant, NF/P = number of fruits per plant, FL = fruit length, FD = fruit diameter, AFW = average fruit weight, Y/P= yield per plant.

**Table 2 plants-10-01860-t002:** Characteristics of the 30 investigated SSR markers and the diversity detected in 51 bitter gourd accessions.

S. No.	Primer Name	Primer Sequence	T_a_ (°C)	A_o_	F_X_	Ge	H_e_	H_o_	PIC	F
1	AVRDC BG-66	F:AGAGGTCTGCCTCTTCCAAA R:CAAGGAACGCAGAAATCCTA	50.0	5	0.38	7	0.73	0.10	0.682	0.87
2	AVRDC BG-2	F:GAGCACACAGAAAATTGGGT R:TGATCCACTCCCAATCTTAGC	51.0	2	0.90	3	0.19	0.02	0.174	0.90
3	AVRDC BG-1	F:CAAGGAACGCAGAAATCCTA R:GAGGTCTGCCTCTTCCAAAA	50.0	3	0.51	3	0.52	0.00	0.403	1.00
4	AVRDC BG-83	F:TATGCAGGGAAGACTGATGG R:TTTTGCTGGCTAAGGTGTTG	50.0	4	0.87	6	0.25	0.08	0.233	0.69
5	AVRDC BG-74	F:AACACCTTCTGACTCCACCC R:CGTTCAATCCTCTCCTCCTC	52.0	2	0.79	3	0.34	0.04	0.281	0.89
6	BG_SSR-8	F:TTAGCTCGCAGTCGCCAC R:ATT GGC CTCAGCAGCCAG	52.0	2	0.98	2	0.04	0.00	0.038	1.00
7	AVRDC BG-95	F:GTTCTCAATTGCATCCGCTA R:CAGCAACAGCAACAGCAGTA	50.0	3	0.59	3	0.50	0.00	0.393	1.00
8	AVRDC BG-7	F:CAGAATCATTGAGAGTGCCG R:GGCCGTAAGCTCTCACACTC	52.0	3	0.83	3	0.30	0.08	0.274	0.74
9	McSSR-20	F:GGAATTCAGGTGAACCTGACG R:CCAGGAGGAAGAGGAACTGC	54.0	2	0.61	2	0.48	0.00	0.363	1.00
10	McSSR-22	F:CCATGACCGATGTAGCACTCC R:TCGAACCAACCTAAACCAG	54.0	2	0.88	2	0.21	0.24	0.186	0.00
11	S13	F:TTGGTTGTGGTGCTGAGTTC R:GATGTAGGGGTTGGGTTGAT	58.0	4	0.65	5	0.52	0.04	0.465	0.93
12	S33	F:ATTTAGTGGGGCGGGTAGT R:TGGATGAGCATGTTAGGGATC	58.0	3	0.57	4	0.56	0.04	0.476	0.93
13	JY 003	F:GTGGGTGCAATGGGTGTC R:CTGCTGCTGTTGCTTCTTC	56.0	5	0.43	6	0.72	0.10	0.673	0.97
14	JY004	F:GTCAACTGCCATCGGTAC R:AGGGAAGAAGAAGAAGAAG	56.0	4	0.57	4	0.61	0.00	0.562	1.00
15	JY006	F:TTTCCAGAGGAGCAGA R:GCTCAGAACTGGCACA	56.0	3	0.45	3	0.65	0.00	0.573	1.00
16	JY009	F:TAAACAACAAAACCAC R:CTCAGAGTCAGAGCAA	52.0	4	0.52	5	0.60	0.10	0.529	0.98
17	JY011	F:AAGTTGGGTTTACGAGTG R:TGGATGATGTAGGGTTTC	52.0	4	0.47	7	0.64	0.10	0.572	0.85
18	S9	F:TTCCCATTCACAGATCACTCC R:CCACCAAATTCAAGAACCCAC	58.0	3	0.66	4	0.64	0.10	0.457	0.96
19	N1	F:GTCTTCCAGGTTGGGAACAG R:ATCTGGTTCCTCGGGAGATT	58.0	3	0.66	4	0.64	0.06	0.438	0.88
20	N6	F:GGGAATTCTCAAAGAGCCAGA R:TGGCACACTCTGCATGAAAT	58.0	3	0.88	3	0.64	0.00	0.202	1.00
21	S12	F:GACATCCTTCTTGCCTCTTACA R:GAAACGGAACGAAACCTCA	58.0	3	0.43	4	0.61	0.04	0.526	0.94
22	JY001	F:GGCTCAGAACTGGCACAG R:TATCACCCATCCATTCAC	56.0	3	0.35	4	0.67	0.10	0.593	0.97
23	JY005	F:TTTATAGCAAACGGCTCA R:GAACATATCGCAAACCTTA	56.0	3	0.42	3	0.65	0.00	0.570	1.00
24	JY008	F:CTCGAACTTTCTGCTC R:TGAATTGAATTGCTCT	54.0	3	0.66	4	0.47	0.10	0.384	0.96
25	N9	F:ATCCATCCCCACAAGTTGAA R:CCATAAGGATATGTTTGCATGG	56.0	3	0.79	3	0.35	0.00	0.298	1.00
26	S18	F:TATGGGTTTTTTCCCCCTCTT R:CATCCCCACAAGTTGAAGAA	56.0	3	0.74	4	0.42	0.10	0.381	0.95
27	S24	F:GCTCTGCGTTTCATTCTTCA R:TGAACCCTCAGACTCAAACTC	56.0	5	0.91	8	0.76	0.08	0.721	0.90
28	S26	F:GAACGCCCTGTGACTTTAGC R:TTTCGTCTTCCAATGAGCC	57.0	4	0.61	4	0.58	0.00	0.533	1.00
29	S20	F:CCCCTTCTAATCACAACCAA R:GGCCTAATTTCTGCCCTTT	57.0	3	0.79	3	0.36	0.00	0.322	1.00
30	S32	F:CTAAATCACGCAAACCCATC R:GAGCAAAAGACTGAGGAAAACT	56.0	5	0.57	6	0.61	0.04	0.570	0.94
Mean			54.7	3.3	0.65	4.1	0.51	0.05	0.429	0.94

A_o_ = Number of alleles per locus, F_x_ = major allele frequency, G_e_ = the number of accessions involved, H_e_ = expected heterozygosity/gene diversity, H_o_ = observed heterozygosity, PIC = polymorphism information content, F = fixation index (inbreeding-like effects within the entire population).

**Table 3 plants-10-01860-t003:** List of bitter gourd accessions used in the study.

No.	Accessions Name	Source of Collection	Salient Features
1	Sel-30-2	Philippines	Light green, medium-long fruits with discontinuous ridges, medium vine
2	Sel-57	China	Dark green, medium-long fruits with smooth tubercles, discontinuous ridges, medium vine
3	DBGS-33	West Bengal, India	Dark green, extra-long fruits with discontinuous ridges, medium vine
4	DBGS-7	IARI, New Delhi	Whitish green, medium-long fruits with continuous ridges, medium vine
5	DBGS-52	West Bengal, India	Dark green, extra-long fruits with discontinuous ridges, medium vine
6	Sel-30-1	China	Light green, small fruits with discontinuous ridges, medium vine
7	Sel-32	Philippines	Dark green, long fruits with continuous smooth ridges, medium vine
8	Sel-59	China	Dark green, extra-long glossy fruits with continuous smooth ridges, vigorous long vine
9	Sel-37	Pakistan	Dark green, extra-long fruits with continuous ridges, medium vine
10	Sel-58	China	Light green, long fruits with continuous smooth ridges, medium vine
11	Sel-2	IARI, New Delhi	Dark green, extra-long fruits with continuous sharp ridges, large long vine
12	Sel-51	Taiwan	Light green, extra-long fruits with discontinuous ridges, large long vine
13	PVGy-1	IARI, New Delhi	Dark green, medium-long fruits with discontinuous ridges, medium vine
14	Sel-33-2	Philippines	Dark green, medium-long fruits with discontinuous ridges, medium vine
15	Sel-25	Thailand	Dark green, small ovate fruits with discontinuous ridges, sharp tubercles, short vine
16	Sel-41	Bangladesh	Dark green, long fruits with continuous ridges, medium vine
17	Sel-26	Thailand	Dark green, long fruits with discontinuous ridges, medium vine
18	NEH-4	Manipur, India	Dark green, medium-long fruits with discontinuous ridges, medium vine
19	Pusa Aushadhi	IARI, New Delhi	Light green, medium-long glossy fruits with continuous narrow ridges, medium vine
20	DBGS-42	West Bengal, India	Light green, medium-long fruits with discontinuous ridges, medium vine
21	NEH-1	Manipur, India	Dark green, medium-long fruits with discontinuous ridges, medium vine
22	Sel-53	Taiwan	Light green, long fruits with continuous smooth ridges, medium vine
23	DBGS-3	Odisha, India	White, extra-long fruits with continuous ridges, vigorous vine
24	Sel-1	IARI, New Delhi	Whitish green, long fruits with discontinuous ridges, medium vine
25	DBGS-9	Odisha, India	Light green, long fruits with continuous ridges, medium vine
26	Arka Harit	IIHR, Bangluru	Dark green fruits with discontinuous ridges, vigorous long vine
27	Pusa Do Mausumi	IARI, New Delhi	Fruits glossy, green with continuous ridges, vigorous long vine
28	Pusa Vishes	IARI, New Delhi	Dark green, long fruits with discontinuous ridges, medium vine
29	Sel-46	Taiwan	Dark green, long fruits with continuous smooth ridges, medium vine
30	Sel-38	Lao	Green, medium-long fruits with discontinuous ridges, medium vine
31	DBGS-52	West Bengal, India	Dark green, small fruits with discontinuous ridges, medium vine
32	MC-84	Kerala, India	Dark green, long fruits with discontinuous ridges, medium vine
33	Pusa Poorvi	IARI, New Delhi	Attractive dark green, small fruits with discontinuous ridges, pointed tubercles, short vine
34	Sel-52	Taiwan	Dark green, long fruits with discontinuous ridges, medium vine
35	NEH-2	Meghalaya, India	Dark green, medium-long fruits with discontinuous ridges, medium vine
36	DBGS-4	IARI, New Delhi	White, long fruits with continuous ridges, medium vine
37	Sel-52-3	Taiwan	Pearl white, medium fruits with discontinuous ridges, medium vine
38	Pusa Rasdar	IARI, New Delhi	Dark green, capsicum shaped, medium with continuous smooth ridges, medium vine
39	PDMGy-201	IARI, New Delhi	Dark glossy green, medium fruits with continuous ridges, medium vine
40	DBGS-33-2	West Bengal, India	Green, medium fruits with discontinuous ridges, medium vine
41	Andhra Collection	Andhra Pradesh, India	Dark green, medium-long fruits with discontinuous ridges, medium vine
42	Nakhra Local	Odisha, India	Medium vine, long, dark green fruits with continuous ridges
43	Sel-33	Philippines	Dark green, medium fruits with discontinuous ridges, medium vine
44	Sel-27	Taiwan	Dark green, medium-long fruits, pointed tubercles with discontinuous ridges, medium vine
45	DBGS-38-1	West Bengal, India	Dark green, small fruits, tubercles presence with discontinuous ridges, medium vine
46	NEH-3	Manipur, India	Dark green, medium-long fruits with discontinuous ridges, medium vine
47	DBGS-41	West Bengal, India	Green, medium-long fruits with continuous smooth ridges, medium vine
48	Sel-52-1	Taiwan	Whitish green, long fruits with continuous ridges, medium vine
49	Sel-52-2	Taiwan	White, long fruits with continuous ridges, medium vine
50	Sel-29	Philippines	Green, medium fruits with discontinuous ridges, pointed tubercles, medium vine
51	DBGS-46	West Bengal, India	Green, long fruits with continuous ridges, medium vine

## Data Availability

Not applicable.

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
