# Peer review of "Analysis of Genetic Diversity and Population Structure in Bitter Gourd (Momordica charantia L.) Using Morphological and SSR Markers"

_plants, 2021, doi:10.3390/plants10091860_

Round 1

Reviewer 1 Report

The submitted paper by the title "Analysis of genetic diversity and population structure in bitter gourd (Momordica charantia L.) genotypes using morphological and SSR markers" shows meaningful information genetic diversity, and relationship between quantitative traits and SSR markers by clustering in bitter gourd. I have some of questions as follows:
1. Lines 293-303, 307-310: You should describe the difference of accessions between the 51 accessions/genotypes used in this study and those of reference no. 24 (lines 514 to 517), also SSRs. The papers used SSR analysis.
2. Lines 225-227: More explanation is needed results about Figure 5.
3. Line 358: Table 1 is arranged followed by Table 3. Please, check Table arrangement.
4. Minor revision was checked on the manuscript. 

Author Response

Comment 1. Lines 293-303, 307-310: You should describe the difference of accessions between the 51 accessions/genotypes used in this study and those of reference no. 24 (lines 514 to 517), also SSRs. The papers used SSR analysis.

Reply: The SSR markers used for genetic diversity in this study were different from those used in our study. Saxena et al. [24] used 54 bitter gourd accessions which were quite different and represent only a few states of India and mostly exotic collections but in our study, diverse origin accessions including India were used. These accessions include gynoecious lines and commercially cultivated varieties of India. Therefore, the polymorphic SSR markers of the present study will be highly useful in fingerprinting, genetic mapping and association analysis in bitter gourd for several economic traits.

Comment 2. Lines 225-227: More explanation is needed results about Figure 5.

Reply: Now Figure 5 is Figure 4 because the other reviewer suggested deleting NJ tree (Figure 4).

The explanation of Figure 4 is added in text as follow:  

Principal Component Analysis (PCA) data for all 51 accessions are shown in Figure 4. The analysis classified these accessions into different groups involving different bitter gourd accessions of diverse origin. Among these, the accessions of four groups were highly diverse from rest of the accessions. The accessions of these four groups are, Group-I (Arka Harit, Sel-52 and Nakhra Local), Group-II (Sel-25, Sel-41 and NEH-1), Group-III (Sel-52-1, Sel-52-2, Sel-52-3, Sel-33 and Sel-46), Group-IV (Sel-53, Sel-37, Sel-42, Sel-26, Sel-33-2 and Pusa Aushadhi). The amount of variance accounted for by PCA plot is 13.48% of axis 1, 7.23% of axis 2 and 6.76% of axis 3 with a total of 27.47% for the three axises. This is an acceptable fit, given the good amount of variability from 51 accessions and SSR alleles used in the analysis.

Comment 3. Line 358: Table 1 is arranged followed by Table 3. Please, check Table arrangement.

Reply: Yes arranged as suggested. The

Comment 4. Minor revision was checked on the manuscript.

Reply: All minor corrections suggested in attached PDF have been done.

Reviewer 2 Report

To accept the article, the authors should take into account a number of comments.

First, the Authors incorrectly use the term “genotype”. This term refers to the genetic structure of organisms, that is, it is applicable not to the organisms themselves, but to their genomes, sets of genes and alleles. Therefore, in most cases (LINES 23,29,30,35,78, 81,82,83,85,93,95,99,108,136-149, 150,154,204,214-220 and below), the word “genotype” should be replaced by “accession” or ”line” when it comes to the studied lines or “specimen” when it comes to individual plants. Unfortunately, a similar error occurs in some publications of other authors.

The Title of the paper should be changed to: Analysis of genetic diversity and population structure of bitter  gourd (Momordica charantia L.) using morphological and SSR markers.

Table 2 is best presented as supplementary material.

Figures 1 and 2 should be compressed horizontally, the distance between the branches should be increased vertically and the font should be enlarged for the accessions names.

Figure 3 does not show data for K = 2, while the text indicates that this value is optimal. The right side of this figure is not explained and is not mentioned in the text.

Figure 4 contains no information other than Figure 3 and should be deleted.

The phrase “The similar results were obtained with UPGMA tree as 4 small fruited genotypes (Nakhra Local, Arka Harit, DBGS-7 and Sel-30-2) were clustered in lower-left side branches of the UPGMA tree (Figure 4), whereas the rest of the genotypes have been clustered in other side of the UPGMA tree”(Lines 218-221) is incomprehensible and should be deleted.

The source of the structure of primers used in the work is nowhere indicated in the paper.

LINES 60-62: “Genetic diversity studies based on morphological characters have been extensively 60 carried out in bitter gourd but based only on morphological characters is no longer sufficient because of low polymorphism” > “Diversity studies based on morphological characters have been extensively carried out in bitter gourd but is no longer sufficient because of low phenotypic polymorphism”

LINE 69: “simple sequence repeat (SSR) is”> “simple sequence repeats (SSRs) are”

LINE 76: delete “genotypes”

LINE 85: Table2 – however Table 1 is absent here. It is at the Methods section and must be renumbered as Table 4.

LINE 108: “Perse performance of 51 genotypes of Momordica species” > “performance of 51 accessions of Momordica charantia”

LINE 137: “by UPGMA method” >” by Unweighted Pair Group Method with Arithmetic Mean (UPGMA)

LINE150-151: delete “Unweighted Pair Group Method of Arithmetic Mean (“

LINE 154:”61 SSR markers” – however there are only 30 investigated markers. The summary states that the rest of the markers did not show polymorphism, but this should also be noted in section 2.3. It makes sense to list the sequences of non-polymorphic markers in the Supplement.

LINE 160: PIC –add “polymorphic information content”

LINE 205: add “actual number of sub-population K [  ]”

LINE 211: delete “Unweighted Pair Group Method of Arithmetic Mean (“

LINES 224-226 The correspondence and similarity of results obtained by UPGMA, STRUCTURE and PCA is not evident.

LINES 279-280: “The variety Pusa Rasdar was also diverse from remaining 10 genotypes in cluster-II” Pusa Aushadi is close to Pusa Rasdar

LINE 299 “They estimated” – Who are they?

LINE 311, 318: delete “Polymorphic Information Content”

LINE 318: “whereas [22] found” >: “whereas Dalamu et al. [22] found”

LINE 320: “In this study” –in our study or study of Dalamu?

LINE 322: “In the current” > “In the current study”

LINES 384-387: 2+1+1+1+2+0.2 mkl is not equal to 20mkl

LINE 433: ref [42] describes MEGA, give ref. or URL for NTSYS; [43] describes NJ method,

It is not clear whether the tree was built with the NTSYS or MEGA programs.

Lines 437-438: “Unweighted Pair Group Method with Arithmetic Mean (UPGMA) tree was also drawn” –it is a repeat of a text above

Line 544: Correct fonts.

Author Response

Comment 1. First, the Authors incorrectly use the term “genotype”. This term refers to the genetic structure of organisms, that is, it is applicable not to the organisms themselves, but to their genomes, sets of genes, and alleles. Therefore, in most cases (LINES 23,29,30,35,78, 81,82,83,85,93,95,99,108,136-149, 150,154,204,214-220 and below), the word “genotype” should be replaced by “accession” or ”line” when it comes to the studied lines or “specimen” when it comes to individual plants. Unfortunately, a similar error occurs in some publications of other authors.

Reply: The term genotype has been replaced with accession wherever it is required as suggested by the esteemed reviewer.

Comment 2. The Title of the paper should be changed to Analysis of genetic diversity and population structure of bitter gourd (Momordica charantia L.) using morphological and SSR markers.

Reply: Title has been changed as suggested.

Comment 3. Table 2 is best presented as supplementary material.

Reply: Table 2 is arranged as Table 1 which was suggested by another reviewer. Table 1 represents the quantitative data for earliness and yield traits which were discussed well in the manuscript by correlating with the SSR results. Therefore, authors want that table 1 should be the part of manuscript not as a Supplementary table. 

Comment 4. Figures 1 and 2 should be compressed horizontally, the distance between the branches should be increased vertically and the font should be enlarged for the accessions names.

Reply: Excellent suggestions by the esteemed reviewer. We think it should be done at the editing stage by the editorial team depending on the space allowed for Figure 1 and Figure 2.

Comment 5. Figure 3 does not show data for K = 2, while the text indicates that this value is optimal. The right side of this figure is not explained and is not mentioned in the text.

Reply: Actually K=3 because the entire genotypes were divided into three groups (Figure 2). The same has been incorporated in the text. The right side figure (Figure 2) is also explained in the manuscript text.

Comment 6. Figure 4 contains no information other than Figure 3 and should be deleted

Reply: Yes deleted.

Comment 7. The phrase “The similar results were obtained with UPGMA tree as 4 small fruited genotypes (Nakhra Local, Arka Harit, DBGS-7 and Sel-30-2) were clustered in lower-left side branches of the UPGMA tree (Figure 4), whereas the rest of the genotypes have been clustered in other side of the UPGMA tree”(Lines 218-221) is incomprehensible and should be deleted.

Reply: Yes deleted

Comment 8. The source of the structure of primers used in the work is nowhere indicated in the paper.

Reply: The source of SSR primers were included in the text as follows: The SSR markers used in the present study includes McSSR series [24], AVRDC-BG series [47], JY series [48], N and S series [32], A and C series [23], MC series [25] and BG_SSR series.
Comment 9. LINES 60-62: “Genetic diversity studies based on morphological characters have been extensively 60 carried out in bitter gourd but based only on morphological characters is no longer sufficient because of low polymorphism” > “Diversity studies based on morphological characters have been extensively carried out in bitter gourd but is no longer sufficient because of low phenotypic polymorphism”

Reply: Done as suggested in text.
Comment 10. LINE 69: “simple sequence repeat (SSR) is”> “simple sequence repeats (SSRs) are”
Reply: Corrected as suggested.
Comment 11. LINE 76: delete “genotypes”
Reply: Deleted in text.
Comment 12. LINE 85: Table2–however Table 1 is absent here. It is at the Methods section and must be renumbered as Table 4.

Reply: Re-arranged tables.

Comment 13. LINE 108: “Perse performance of 51 genotypes of Momordica species” > “performance of 51 accessions of Momordica charantia”

Reply: Corrected.

Comment 14. LINE 137: “by UPGMA method” >” by Unweighted Pair Group Method with Arithmetic Mean (UPGMA)

Reply: Corrected.

Comment 15. LINE150-151: delete “Unweighted Pair Group Method of Arithmetic Mean (“

Reply: Deleted

Comment 16. LINE 154:”61 SSR markers” – however, there are only 30 investigated markers. The summary states that the rest of the markers did not show polymorphism, but this should also be noted in section 2.3.

Reply: Yes added in the text as follows: All the 51 accessions were screened using 61 SSR markers but only 30 were polymorphic and produced a total of 99 alleles.

Comment 17. LINE 160: PIC –add “polymorphic information content”

Reply: Added.

Comment 18.   LINE 205: add “actual number of sub-population K [  ]”

Reply: Added.

Comment 19 LINE 211: delete “Unweighted Pair Group Method of Arithmetic Mean (“

Reply: Deleted.

Comment 20. LINES 224-226 The correspondence and similarity of results obtained by UPGMA, STRUCTURE and PCA is not evident.

Reply: Corrected in the text.
Comment 21. LINES 279-280: “The variety Pusa Rasdar was also diverse from remaining 10 genotypes in cluster-II” Pusa Aushadi is close to Pusa Rasdar

Reply: The sentence was re-arranged in the text.

Comment 22.  LINE 299 “They estimated” – Who are they?

Reply: They reefers to Dalamu et al. [24]. Incorporated in text.

Comment 23. LINE 311, 318: delete “Polymorphic Information Content”

Reply: Deleted

Comment 24. LINE 318: “whereas [22] found” >: “whereas Dalamu et al. [22] found”

Reply: Yes corrected.

Comment 25.  LINE 320: “In this study” –in our study or study of Dalamu?

Reply:  In the study of Dalamu. Same incorporated in text

Comment 26 LINE 322: “In the current” > “In the current study”

Reply: Yes corrected

Comment 27. LINES 384-387: 2+1+1+1+2+0.2 mkl is not equal to 20mkl

Reply: Corrected in text as follows: Each microsatellite PCR amplification reaction was performed with 10µL reaction volume containing 2µl of 25ng genomic DNA, 1.0µl of 10mM each of deoxynucleotide triphosphates (dNTPs), 1µl of each of reverse and forward primer, 2µl 1X PCR standard Taq buffer (Tris with 15mM Mgcl2) and 0.2µl of Taq DNA polymerase (0.3U/µl) (Bangalore GeNei Pvt. Ltd., Bengaluru, India) and 2.8µl molecular grade water.  

Comment 28. LINE 433: ref [42] describes MEGA, give ref. or URL for NTSYS; [43] describes NJ method,

Reply:  Described MEGA, given URL for NTSYS and NJ also described.

Comment 29.  It is not clear whether the tree was built with the NTSYS or MEGA programs.

Reply: NTSYS used. Deleted sentence about MEGA

Comment 30. Lines 437-438: “Unweighted Pair Group Method with Arithmetic Mean (UPGMA) tree was also drawn” –it is a repeat of a text above

Reply:  The repeated sentence was deleted.

Comment 31. No need to provide a separate subheading PCA analysis as it is also used for establishing relationships among the variables and finer scale analysis of cluster analysis.

Reply:  The separate sub heading PCA analysis has removed in revised MS

Comment 32. Line 544: Correct fonts.

Reply:  Done

Reviewer 3 Report

the article presented by a group of authors is devoted to the assessment of the genetic diversity of the bitter gourd. The purpose of this work is logical, since India is the main growing area of ​​the bitter gourd. A lot of work has been done, neatly framed, there is no doubt about the results.
There are some small remarks: there is a lot of data in the discussion section, which can be transferred to the results section and leave only the discussion of the results obtained.
table 1 came after table 3
Table 3 presents the summary data of 51 pumpkin genotypes for each marker? Are they the same for all genotypes?

Author Response

Comment: 1. The article presented by a group of authors is devoted to the assessment of the genetic diversity of the bitter gourd. The purpose of this work is logical, since India is the main growing area of ​​the bitter gourd. A lot of work has been done, neatly framed, there is no doubt about the results. There are some small remarks: there is a lot of data in the discussion section, which can be transferred to the results section and leave only the discussion of the results obtained.

Reply: Some of the data presented in discussion part transferred to result section as suggested.

Comment: 2. table 1 came after table 3

Reply: Rearranged.

Comment: 3. Table 3 presents the summary data of 51 pumpkin genotypes for each marker? Are they the same for all genotypes?

Reply: Yes table 3 (now table 1) presents the summary data of 51 bitter gourd genotypes which were correlated with the 30 polymorphic SSR markers.